# The Impact of Mandatory Food-Marketing Regulations on Purchase and Exposure: A Narrative Review

**DOI:** 10.3390/children10081277

**Published:** 2023-07-25

**Authors:** Alanoud Alfraidi, Nora Alafif, Reem Alsukait

**Affiliations:** 1Department of Community Health Sciences, College of Applied Medical Sciences, King Saud University, Riyadh 11433, Saudi Arabia; nalafeef@ksu.edu.sa (N.A.); ralsukait@ksu.edu.sa (R.A.); 2Department of Clinical Dietitian, Prince Sultan Military Medical City, Riyadh 12233, Saudi Arabia

**Keywords:** advertising, marketing, regulation, government policy, health impact, effectiveness

## Abstract

(1) Background: Several governments have enforced a series of actions to improve the local food environment and reduce obesity-related diseases in the population by implementing statutory regulations to reduce or ban the marketing of products that are considered unhealthy based on nutrient profile systems or them being high in fat, sugar, and salt (HFSS); (2) Objective: This narrative review is aiming to provide a comprehensive exploration of the available evidence on the impact of identified mandatory regulations restricting food marketing, including advertisements and packages on the exposure and purchase of HFSS food products, to help justify the need for these regulations; (3) Methods: Articles were retrieved by searching electronic databases, including EBSCO Education, PubMed, Scopus, Web of Science, and Google Scholar from 2012 up to December 2022; (4) Results: A total of 12 articles were included in this review. Almost all mandatory food-marketing regulations have evidence in favor of reducing HFSS food purchases and exposure; (5) Conclusions: Protecting children and adolescents from food and beverage marketing through mandatory regulations is a crucial step toward tackling global childhood and adolescent obesity and securing a healthier environment for future generations.

## 1. Introduction

In 2016, the World Health Organization (WHO) reported that over 340 million children and adolescents aged 5–19 were overweight or obese [1]. According to the Organization for Economic Co-operation and Development (OECD), obesity shortens lives by 2.7 years across all OECD nations [2]. The OECD predicts that, over the next three decades, 8.4% of the health budgets of member countries will go toward treating the effects of obesity [2]. In turn, multiple studies have found that digital and social media food marketing has played a part in the increase in adolescent obesity, food behavior, and attitudes toward high-fat, -sugar, and -salt (HFSS) foods [3,4,5,6].

External eating, as proposed by Schachter’s theory on obesity, refers to the eating behavior of individuals who rely on external cues, such as the presence of food, time of day, or sensory cues, rather than the internal physiological cues of hunger and satiety, to determine when to eat [7]. These people are more likely to eat without hunger and consume more food than necessary [7]. Alternatively, psychosomatic theory suggests that obese individuals overeat as a reaction to emotional arousal [7]. Thus, according to this theory, children may interpret negative feelings as hunger and engage in overeating to numb their emotions and reduce anxiety [7]. A cross-sectional study found that 10.5% of overweight children exhibited emotional eating and 38.4% reported external eating, while overweight adolescent females reported high levels of emotional eating and overweight adolescent males displayed more external eating [7].

Advertisers use emotions, memory, attention, perception, reward, and approach and withdrawal motivation processing in the brain to catch consumers’ attention and increase purchase intentions and unconscious behaviors [8]. In addition, pleasure and displeasure can be used to sway customers toward the goal of advertisements [8]. Food advertising aimed at children is typically for HFSS foods to normalize the consumption of such foods and associate them with aroma, taste, magic, fun, humor, physical activity, and exaggerated pleasure [9]. The advertisements used in junk food marketing imply that children will receive emotional benefits from the consumption of junk food, as illustrated by successful adolescent-targeted campaigns such as “Open Happiness” (for sugary soda), “You’re Not You When You’re Hungry” (for a candy bar), and “Win from Within” (for a sports drink) [10]. At the same time, adolescence is a period of neural imbalance caused by early maturation of the brain’s reward system and delayed maturation of the prefrontal cortex (the brain region responsible for behavioral control), resulting in the mature reward system taking the lead [11]. This opposition means that adolescents have difficulty making rational decisions, such as resisting cheap fast-food advertisements when there is the potential for a reward [11]. The insecurity of adolescents compels them to elevate their self-worth by using popular brands and practicing materialism, which makes them as vulnerable as young children to the influence of food marketing [12,13].

Food advertisements targeting children and adolescents bring attention and awareness to the product and create emotional connections that influence attitudes and preferences toward the product [10]. It has been found that an increase in the presence of HFSS food advertising in a child’s environment is associated with higher consumption of these food products [14]. In addition, high advertisement recall was found to cause children to have a positive attitude toward unhealthy foods and to increase their intention to buy these products [15]. Several studies have found social media food marketing to be associated with an increased preference for sweets and unhealthy foods and to discourage adolescents from making healthy food choices [16,17]. Advertisers are also using advanced medical technology to their advantage, employing neuroimaging tools such as fMRI, EEG, and fNIRS to analyze the neural responses associated with consumers’ behaviors, such as decision-making, choices, perception, and preferences, that contribute to their reactions to marketing [18]. Visual attention, emotional arousal, and pleasure/displeasure toward a marketing campaign can also be measured using physiological tools such as ET, GSR, and ECGs [18]. Both neuroimaging and physiological tools are invaluable in capturing consumers’ mental and physiological responses toward marketing and measuring the effectiveness of food-marketing regulations [18].

Increased exposure to the marketing of HFSS food has occurred in tandem with the global childhood obesity epidemic [19]. In a recent cohort study, it was shown that a calorie increase of only 69 to 77 kcals/day is needed to produce an overweight child [20]. Furthermore, several studies have linked digital and social media food marketing with an increase in unhealthy food intake and total calorie intake among children and adolescents [4,5,6,16,21,22]. Exposure to the marketing of HFSS food normalizes its consumption and influences attitudes, purchasing choices, and consumption behavior regarding these products across all age groups, especially children [23]. There is strong and consistent evidence that mandatory governmental regulation is required to protect children from the negative impacts of food marketing, as voluntary regulations are largely ineffective [24,25]. Nevertheless, there needs to be more action internationally, as the existing regulations follow old and ineffective guidelines, involve self-regulation or codes of practice, and rarely consider the impact of social media marketing [24]. Thus, in 2016, the WHO urged countries to regulate the marketing of products high in saturated fats, trans-fatty acids, sugars, and/or salt to children [26].

An ecological study evaluating the impact of junk food broadcast marketing policies on nationwide junk food sales reported that, between 2002 and 2016, countries with mandatory regulations experienced a drop in sales per capita (−8.9%), whereas those with only self-regulatory measures experienced an increase (+1.7%, *p* = 0.004) [25]. Thus, there is general agreement among public health professionals that mandatory regulatory measures are needed to advance obesity prevention policies [27]. Evidence of their effectiveness is necessary to strengthen the argument in favor of these policies. In response, the current review provides a comprehensive exploration of the available evidence on the impact of mandatory food-marketing regulations, including advertisements and packages, on the exposure and purchase of HFSS food products, to help justify the need for these regulations. Section 2 contains a description of the methodology and data collection process used in this study. Section 3 lists the effects of food-marketing regulations on four countries (Chile, UK, Ireland, and Spain) employing such regulations. A discussion of the study’s findings is presented in Section 4, and Section 5 specifies the limitations and suggests future directions and implication of the study. Finally, Section 6 presents the study’s conclusions. This review differs from previous work carried out in the same area by focusing only on mandatory regulations with exposure and purchasing activities as outcomes.

## 2. Materials and Methods

### 2.1. Review Question and Population, Exposure, Comparison, and Outcome (PECO) Statement

We conducted a narrative review with a comprehensive exploration of the available evidence on the impact of identified mandatory regulations restricting food marketing through advertisements and packages on the exposure and purchase of HFSS food products. We used the PECO statement for the search strategy; details are described in Table 1 below.

### 2.2. Search Strategy

Articles were retrieved by searching these bibliographic databases: EBSCO Education, PubMed, Scopus, Web of Science, and Google Scholar from 2012 up to October 2022. The searches were carried out in December 2022. Search terms were combined by Boolean logic (AND, OR). The keyword combinations used included the following keywords: (Effect* OR impact* OR consequence* OR outcome*) (food* OR diet* OR beverage* OR drink*) AND (market* OR advert* OR promotion*) AND (regulated* OR restrict* OR law* OR polic*). The reference lists of the identified articles were manually searched for potentially relevant studies. Duplicated articles were removed manually. Articles were screened first by title and abstract, then by full text. An overview of the search and screening process is provided in Figure 1.

### 2.3. Study Eligibility Criteria

#### 2.3.1. Inclusion Criteria

The inclusion criteria included all study designs, and English-language, peer-reviewed evaluations that examined the impact of implemented mandatory policies to restrict food marketing through advertisements and packages compared with no regulations (e.g., before the regulation was implemented), published from 2012 to December 2022. Critical outcomes were exposure to HFSS food marketing and the purchase of these products.

#### 2.3.2. Exclusion Criteria

Exclusion criteria were studies assessing the impact of policies yet to be implemented, voluntary regulations, or other food-marketing techniques such as product placements in the supermarkets.

### 2.4. Data Extraction

Data extraction included information on the study country, study design, policy, year of implementation, medium, and outcome measures of the impact of the regulations.

## 3. Results

As shown in Figure 1, there were a total of 7643 papers, out of which there were 7609 screened studies and 12 selected for final inclusion in our review. The studies were reviewed by all authors to ensure they met the inclusion and exclusion criteria.

### 3.1. Characteristics of Included Articles

A total of 12 studies were selected for a complete review. The oldest study was published in 2012 and the most recent in 2021. Six articles were carried out in Chile, one in Ireland, one in Spain, and four in the UK. Ten studies assessed the impact of policies on food-marketing exposure and two studies assessed the impact on purchase. The regulations in the selected studies protected children under 14, under 16, and under 18 years old from unhealthy food marketing. The study designs used in the studies were repeated cross-sectional and time series designs. The most common regulations and restrictions in these studies were on television advertising during children’s programs, and schools were the common setting. Regulations on new media such as social media, cinema, mobile phone applications, packaging, and the Internet were uncommon. These studies are described in Table 2.

In the Chile studies, one evaluated the impact of Chile’s law on food labeling and advertising on sugar-sweetened beverage purchases, four evaluated the exposure to high-in food marketing with child-directed appeals, and the last one evaluated the exposure to high-in food marketing with child-directed appeals on breakfast cereal packages. In the UK studies, three studies evaluated the exposure of child-directed TV food advertising post-regulation, and the other study evaluated the impact of food advertisement restrictions across transport on purchase. Spain’s study evaluated the exposure of HFSS in TV advertisements on children’s channels compared with general channels post-regulation. Similarly, Ireland’s study evaluated the exposure of HFSS post-regulation on children’s channels compared with general channels. More details are shown in Table 3.

### 3.2. Mandatory Regulations Evaluation Studies

#### 3.2.1. Chile

Several studies have been carried out to evaluate Chile’s regulations. Firstly, purchases of the “high-in” products that exceeded nutrient thresholds (i.e., were subject to policy restrictions) declined, while there was an increase in “not-high-in” nutrient purchases [31]. In addition, the overall calories purchased declined by 4%, the overall purchase of sugar declined by 10%, the purchase of saturated fat declined by 4%, and the purchase of sodium declined by 5% [31]. Secondly, exposure to high-in food advertising in total decreased significantly by 44% among preschoolers and 58% among adolescents; this decrease occurred in programs meant for children (*p* < 0.001) as well as general (*p* < 0.001) [28,29,30,33]. Thirdly, exposure to high-in food marketing with child-directed appeals, such as cartoon characters, declined by 35% among preschoolers and 52% among adolescents [28,32]. Sugar, which was the most prevalent nutrient in high-in marketing seen by adolescents before the Chilean regulations, decreased by 60%, followed by calories, which decreased by 68%, while saturated fats showed the highest decrease at 72% [28,30]. Lastly, the use of child-directed marketing strategies in Chilean breakfast cereal packages that specifically used a character declined from 30% to 21% post-implementation, but this difference was not statistically significant (*p* = 0.07) [32]. Nevertheless, among those packages that included characters, the overall percentage of products using licensed characters dropped considerably from 13% to 0.8% after implementation, as did the percentage of packages including physically active characters (*p* < 0.05) [32]. When comparing “high-in” packages, the prevalence of those with at least one child-directed strategy significantly decreased from before to after implementation (*p* < 0.05), and the use of characters decreased significantly as well, from 36% of “high-in” packages before implementation to 15% of “high-in” packages after implementation (*p* < 0.05) [32]. Additionally, there was a considerable drop in the use of non-character techniques, which went from 23% prior to implementation to 0% afterward (*p* < 0.05) [32].

#### 3.2.2. The United Kingdom

The evaluation of the UK regulation in 2010 regarding children’s exposure to HFSS food advertising reported that the exposure did not change between 6 months before and 6 months after [34]. This is probably because the UK only implemented the policy for a few television broadcasts [34]. The other studies in the UK reported a reduction in total TV HFSS advertising exposure post-regulation [35,36]. Nevertheless, other media partially compensated this decline and increased food advertisements by 4.7% from non-peak children’s viewing times, with a higher number of non-core foods advertised at these times (+0.5%) [36]. On the other hand, the UK reported that energy purchases from HFSS products were 6.7% lower, and it observed a relative reduction in purchases of fat, saturated fat, and sugar from HFSS products after restrictions on the advertisement of HFSS products across the Transport for London network [37].

#### 3.2.3. Ireland

According to post-regulation research in the Republic of Ireland, children’s networks like Nickelodeon and Nick Junior featured one food advertisement every two hours, whereas general commercial channels showed one every 10–15 min [9]. Many children view general commercial channels, so their exposure to HFSS food advertisements will increase [9].

#### 3.2.4. Spain

In Spain, the frequency of food advertisements was 19 advertisements per hour (adv/h) on children’s channels and 25 adv/h on the general channels post-regulation [38]. The fast-food advertisements were shown 72 times on the general channels, and the non-core food advertisements were slightly higher than on children’s channels (*p* < 0.001) [38].

## 4. Discussion

All studies that were included in this review have evidence in favor of reducing HFSS food purchases and exposure post regulation implementation [9,28,29,30,31,33,35,36,37,39], except for one study in the UK [34]. Several studies could not distinguish the impacts of labeling, marketing, and school-sale ban rules, nor could they tell whether the observed changes in purchases were the result of consumer behavioral change or reformulation, but they generally decreased exposure and purchases [32,33,40]. Almost all studies did not measure or include the purchases and the exposure that happened outside the household [9,28,29,30,31,33,35,36,37,39]. However, most of the studies had no evidence to propose that the reduction in advertising exposure interposes a reduction in the consumption of HFSS. Nevertheless, two studies reported that adolescents with lower levels of advertisement exposure at baseline reduced their intake of high-in foods (*p* = 0.03), while, in preschool children, high-in food consumption significantly declined (*p* < 0.01) [29,30].

The regulation of child-directed strategies can act as a limitation, as its focus on child-directed strategies is only to be banned by regulations, while other known marketing strategies can be appealing to children, such as images of adolescents or teens and design techniques [9]. Along the same line, marketing regulations on only children’s television programs are also limited in their application, as they focus exclusively on children’s programs, which does not reflect children’s actual viewing time or exposure [9]. The highest viewed programs by children were reality shows and sporting events, even though they are aimed at the general population, and the ads during such shows might captivate children’s attention without targeting them [9]. Therefore, the impact of regulations could be underestimated since the exposure to food marketing was only measured based on child programs’ exposure, while children and adolescents could still be exposed to and influenced by food marketing during non-child programs.

The Net Children Go Mobile study in six European countries stated that smartphones were the devices most frequently used daily by children aged 9–16 years old [41]. Children view influencers or vloggers on platforms such as YouTube as authentic, and they trust their recommendations more than brand advertising. Therefore, brands seek mentions from these influencers and vloggers to promote their products [26]. However, most regulations were limited to broadcast and TV advertising and did not address social media, websites, influencers, or product placement within programs. Therefore, regulations should cover all media, including digital, to minimize marketing shifts to other less regulated media platforms.

## 5. Limitations, Strengths, and Future Directions

### 5.1. Limitations

One of the limitations of this review is that we could not include Middle Eastern and African countries since they still need to put mandatory food-marketing regulations in place to the best of our knowledge. Also, regarding the other countries with mandatory regulations that were not included in this review, we could not find studies that evaluated post-implementation regulation in these countries. Chile has the most comprehensive mandatory regulations among other countries. Therefore, the majority of the studies that were included in this narrative review were from Chile, which makes the impact of mandatory regulations hard to generalize, but it can act as a starting point to strengthen the scientific evidence of the importance of mandatory regulations rather than voluntary or industry codes.

### 5.2. Strengths

The strengths of this review are that we focused on the impact of mandatory regulations and were able to identify the barriers to the implementation of regulations, which included legal enforcement guarantees, poor intersectoral collaboration, weakness in scientific criteria, and poor monitoring. In addition, the regulations’ limitations were that they focused on food marketing on TV and radio targeting children, and focused solely on child-directed marketing. In addition, policies did not include new media marketing, such as social media. These findings could help researchers and policymakers develop future policies and limit the barriers to implementation. This review proves the mandatory regulations’ positive impact on reducing the exposure and purchase of HFSS food products and justifies the need for these regulations.

### 5.3. Future Directions and Implication of the Study

Future studies need to explore the possible longer-term effects of HFSS advertising restriction regulations. They also should explore the impact of food-marketing restrictions on outside-the-house purchases and exposure to HFSS products, which may result in even further impacts. Studies need to measure the impacts of the regulations on the consumption of these HFSS foods and quantify the potential impacts of such regulations on obesity and related diseases. As there is no standardized method to classify foods as “unhealthy” or “healthy”, it is difficult to regulate policies regarding the health status of foods. Therefore, adopting a nutrient profiling system (NPS) is a crucial step before implementing regulations, since countries adopting standardized nutrition criteria were associated with a higher decline in sales per capita (−8.6%) of unhealthy food compared with countries without an NPS [25].

Food-marketing regulations should be adapted to account for children’s viewing habits, as children watch family and general television programs in addition to child programs. This should include children’s exposure to digital media, influencers, and vloggers, where current restrictions on marketing should be extended. Most importantly, there needs to be monitoring of how brands and advertisers adapt to the regulations in order to design future regulations.

Governments are gradually implementing laws prohibiting the marketing of HFSS foods to minors, and all such regulations have been in effect since 2007. Meanwhile, current regulations vary regarding children’s ages, media used, type of foods included, and marketing techniques. It is crucial to keep in mind all the neuro-marketing tools and techniques that have been used in marketing to influence the consumer’s behavior when implanting a new regulation [18]. These findings could help researchers and policymakers to adjust loopholes in current policies and help developing future regulations.

## 6. Conclusions

The main objectives of regulations are to encourage changes in eating behaviors by limiting direct exposure to unhealthy foods while indirectly improving food environments and utilizing a positive impact on health [42]. To limit exposure to targeted junk food marketing aimed at children and adolescents effectively, governments should implement strong, comprehensive mandatory regulations and expand the restrictions to cover social media, online games, and peer-to-peer marketing that are directed or not directed at children and adolescents [25,42].

This review proves the mandatory regulations’ positive impact on reducing the exposure and purchase of HFSS food products and justifies the need for these regulations. Protecting children and adolescents from food and beverage marketing through mandatory regulations is crucial in tackling global childhood and adolescent obesity and securing a healthier environment for future generations.

## Figures and Tables

**Figure 1 children-10-01277-f001:**
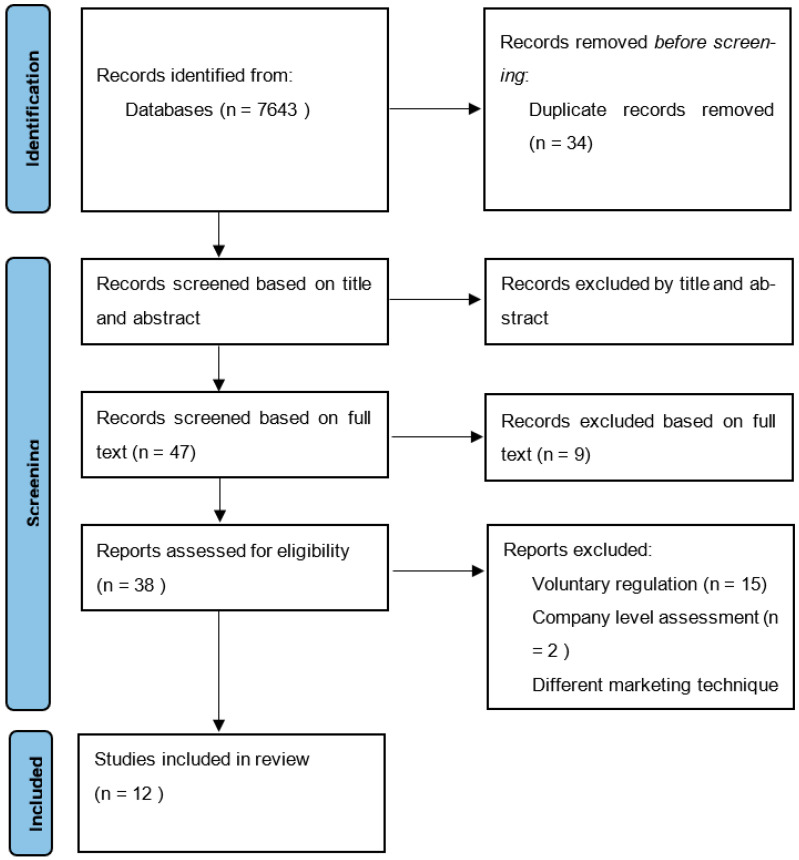
Flowchart of screening and selection procedure of studies for the review.

**Table 1 children-10-01277-t001:** The PECO statement used for the search strategy.

Acronym	Definition	Description
P	Population	All population
E	Exposure	Mandatory policies restricting food marketing through advertisements and packages
C	Comparison	No regulations
O	Outcome	Purchase and exposure

**Table 2 children-10-01277-t002:** Articles included in the review.

References	Country	Policy Evaluated	Year of Implementation	Study Design	Medium	Outcome Reported
[28]	Chile	Chile Food Labeling and Advertising Regulation	2016	Repeated cross-sectional content analysis	TV ads	Exposure
[29]	Chile	Chile Food Labeling and Advertising Regulation	2016	Repeated cross-sectional content analysis	TV ads	Exposure
[30]	Chile	Chile Food Labeling and Advertising Regulation	2016	Repeated cross-sectional content analysis	TV ads	Exposure
[31]	Chile	Chile Food Labeling and Advertising Regulation	2016	Repeated cross-sectional content analysis	Packaging	Purchase
[32]	Chile	Chile Food Labeling and Advertising Regulation	2016	Repeated CS content analysis	Packaging Ads	Exposure
[33]	Chile	Chile Food Labeling and Advertising Regulation	2016	Repeated CS content analysis	TV ads	Exposure
[34]	United Kingdom	UK Code of Broadcast Advertising	2008	Repeated CS content analysis	TV ads	Exposure
[35]	United Kingdom	UK Code of Broadcast Advertising	2010	Repeated CS content analysis	TV ads	Exposure
[36]	United Kingdom	UK Code of Broadcast Advertising	2010	Repeated CS content analysis	TV ads	Exposure
[37]	United Kingdom	UK Code of Broadcast Advertising	2018	Controlled interrupted time series design	Transport ads	Purchase
[9]	Ireland	Advertising Standards Authority for Ireland	2012	Repeated CS content analysis	TV ads	Exposure
[38]	Spain	European and Spanish Public Health laws	2011	Repeated CS content analysis	TV ads	Exposure

**Table 3 children-10-01277-t003:** Food-marketing regulations’ impact.

Country	Policy Area	Sub-Policy Area	Policy Action	Topics	Age	Impact
Chile	Restricts food advertising and other forms of commercial promotion	Mandatory regulation of broadcast food advertising to children	- The law restricts advertising directed to children of food in the “high in” category, including TV programs, the Internet, radio, and magazines. - Promotional strategies such as cartoons are banned, as is advertising in schools.	Advertising, Children, Digital marketing, Marketing, Saturated fat, Sugar	<14 years old	- Calories, sugar, saturated fat, and sodium purchased declined in the “high in” products [31]. - Exposure to high-in food marketing with child-directed appeals declined among preschoolers and adolescents [28,29,30]. - Child-directed marketing decreased significantly post-policy implementation [32]. - Prevalence of child-directed marketing on breakfast cereal packages decreased [32]. - The percentage of marketing of “high in” products decreased post-regulation (*p* < 0.001) [33].
Ireland	Restricts food advertising and other forms of commercial promotion	Mandatory regulation of broadcast food advertising to children	TV advertising to children is prohibited for specific food before, during, and after programs shown up to 6 pm and during other children’s programs.	Advertising, Children, Digital marketing, Marketing	<16	- After the restrictions, the ads during children’s programs were one every 2 h vs. one every 10–15 min during the general programs [9]
United Kingdom	Restricts food advertising and other forms of commercial promotion	Mandatory regulation of broadcast food advertising to children	The 2010 UK Code of Broadcast Advertising (BCAP) Code prohibits advertising and product placement of HFSS food as defined by a nutrient profiling model during and adjacent to TV and radio programs with a particular appeal to viewers (includes sponsorship of TV programs).	Advertising, Children, Marketing, Nutrient profile model	<16	- A reduction in total TV HFSS advertising exposure [35,36]. - Exposure of children to HFSS food advertising did not change before and after the regulation [34].
United Kingdom	Restricts food advertising and other forms of commercial promotion	Mandatory regulation	Transport for London ad policy.	Advertising, Marketing, Nutrient profile model	All age groups	- Energy purchased from HFSS products declined by 6.7%. - A reduction in purchases of fat, saturated fat, and sugar products [37].
Spain	Restricts food advertising and other forms of commercial promotion	Mandatory regulation	European and Spanish Public Health laws.	Advertising, Marketing,	<16	- Exposure to fast food advertisements on general TV was higher than during children’s programs [38].

## Data Availability

Not applicable.

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
