# Peer review of "The Impact of Mandatory Food-Marketing Regulations on Purchase and Exposure: A Narrative Review"

_children, 2023, doi:10.3390/children10081277_

Round 1
Reviewer 1 Report
REVIEW COMMENTS
I have only a few concerns about the paper and some suggestions that maybe the authors could consider:
General comment, as follows:
1. To begin with, there are some typos and grammar mistakes. Some long sentences could make readers confused.
“Introduction” section comments, as follows:
2. In the 'Introduction' sections, the proposed research gap and the stated objectives do not meet the criteria of proper synergy. Please make the research gap and the research objectives consistent with each other.
3. I think the “Introduction” section can be improved. I think the authors should write about consumer behavior (e.g., unconscious and subconscious), which plays a vital role in decision making of consumer toward product. So, I suggest a refs. 'consumer behaviour to be considered in advertising: a systematic analysis and future agenda'. In addition, the authors can enrich “introduction” section with neuroscientific tools that can measured the non-verbal interactions and facial expressions, which are benefits in the study, therefore, I suggest the following reference 'biomedical technology in studying consumers’ subconscious behavior'. I think these references can help you with the issue.
4. I think the author(s) can add a paragraph at the end of the "Introduction" section about the structure of the paper to ease reading for readers and interested persons. I suggest a reference, which can help you in this issue as well as to improve your literature review 'scientometric analysis of scientific literature on neuromarketing tools in advertising'.
5. The authors should explicitly state the novel contributions of this work and its similarities and differences with previous publications.
“Methodology” section concerns, as follows:
6. The methodology section should be clearly clarified due to the fact that there are some unclear issues. I suggest paper which can help the authors to enhance the methodology, as follows: https://doi.org/10.1177/21582440231156563
7. Why do the authors include some countries and ignore others?
8. What kind of methodology that authors follow to extract papers from databases?
“Result” section comments, as follows:
9. The researchers have not adequately explained their methodology or provided sufficient details on the experimental setup, making it difficult to assess the reliability and validity of the results. The lack of transparency raises questions about the rigor of the study. The small number of papers analyzed in a study can be a major weakness, as it may limit the reliability, validity, and generalizability of the findings. Therefore, it is essential to ensure the appropriate methodology and keywords that have been used to extract data. In addition, there is a flaw in the methodology and design of extracting papers. For example, the authors said that the author(s) have extracted data from two databases, then in the next paragraph, the author(s) the data collected from three databases.
“Discussion” section comments, as follows:
10. discussion should be more concise and comprehensive.
General concerns about the missed sections such as “Theoretical and practical implications”, “Conclusion”, “Limitations and Future Directions” in this paper, as follows:
11. The authors need to clearly articulate the academic as well as practical implications of this study in a separate section which can be named the theoretical and practical implications of this study. I suggest paper which can be useful https://doi.org/10.1007/s12144-023-04812-w
12. What are some potential implications of these findings for researchers, psychologists, and governments?
13. The authors need to clearly articulate the limitations and future research of this study in a separate section which can be named ‘limitations and future research’ behind the conclusion section.
14. How does the study contribute to the existing literature on impoliteness in social interaction, particularly in naturally occurring conflicts?
If these revisions can be made to the manuscript, I believe that this study can be accepted for publication.
I wish the authors all the very best with this study.
There are some typos and grammar mistakes. Some long sentences could make readers confused.
Author Response
- To begin with, there are some typos and grammar mistakes. Some long sentences could make readers confused.
We worked on that, hopefully, it's better now.
“Introduction” section comments
We worked on your comments and the references were very useful, thank you.
“Methodology” section concerns, as follows:
6. The methodology section should be clearly clarified due to the fact that there are some unclear issues. I suggest paper which can help the authors to enhance the methodology, as follows: https://doi.org/10.1177/21582440231156563
We worked on that. Please let me know If you have any concerns.
- Why do the authors include some countries and ignore others?
We included only the countries that have mandatory regulation, and it has been evaluated.
What kind of methodology that authors follow to extract papers from databases?
we added it to the methodology part.
“Result” section comments, as follows:
We worked on that
“Discussion” section comments, as follows:
We worked on that
“Conclusion”, “Limitations and Future Directions” in this paper,
We added that, and the reference was useful thank you.
- The authors need to clearly articulate the limitations and future research of this study in a separate section which can be named ‘limitations and future research’ behind the conclusion section.
We added that
Thank you for all your constructive comments.

Reviewer 2 Report
Thank you for allowing me to review this manuscript. The topic is always of interest in scientific research and is timely. The article is well-written but lacks anchoring theory. In the introduction, the authors should describe the literature on the topic, getting to the heart of Health Science theories on this topic. I recommend that the authors revise the introduction. I suggest some references that may be helpful:
- Forman, E.; Butryn, M. A new look at the science of weight control: How acceptance and commitment strategies can address the challenge of self-regulation. Appetite 2015
-Braet, C.; Claus, L.; Goossens, L.; Moens, E.; Van Vlierberghe, L.; Soetens, B. Differences in eating style between overweight and normal-weight youngsters. J. Health Psychol. 2008, 13, 733-743
- GuerriniUsubini, A.; Cattivelli, R.; Bertuzzi, V.; Varallo, G.; Rossi, A.A.; Volpi, C.; Bottacchi, M.; Tamini, S.; De Col, A.; Pietrabissa, G.; et al. The ACTyourCHANGE in Teens Study Protocol: An Acceptance and Commitment Therapy-Based Intervention for Adolescents with Obesity: A Randomized Controlled Trial. Int. J. Environ. Res. Public Health 2021,18,6225. https://doi.org/ 10.3390/ijerph18126225
Related to the review, It needs to be clarified what strategy the authors used to carry out the review. The review must follow specific statements. I would recommend that the authors review all in light of the PRISMA model: http://www.prisma-statement.org
Moreover, the choice of data extraction databases must be justified.
Author Response
Thank you for all your constructive comments. I worked on your comments and thank you for the useful references.
Round 2
Reviewer 1 Report
REVIEW COMMENTS
I have only a few concerns about the paper and some suggestions that maybe the authors could consider:
1. Initially, it is important to address the presence of typographical errors and grammatical inaccuracies within the text. Furthermore, the inclusion of lengthy sentences may result in reader confusion and should be carefully considered for improved clarity and comprehension.
2. I think the “Introduction” section can be improved. I think the authors should write about consumer behavior (e.g., unconscious and subconscious), which plays a vital role in decision making of consumer toward product. So, I suggest a refs. "exploring global trends and future directions in advertising research: a focus on consumer behavior". In addition, the authors can enrich “introduction” section with neuroscientific tools that can measured the non-verbal interactions and facial expressions, which are benefits in the study, therefore, I suggest the following "neuromarketing tools used in the marketing mix: a systematic literature and future research agenda". I think these references can help you with the issue.
3. I think the author(s) can add a paragraph at the end of the "Introduction" section about the organizational structure of the paper to ease reading for readers and interested persons. I suggest a reference, which can help you in this issue as well as to improve your literature review 'scientometric analysis of scientific literature on neuromarketing tools in advertising'.
4. The authors should explicitly state the novel contributions of this work and its similarities and differences with previous publications.
5. The methodology section should be clearly clarified due to the fact that there are some unclear issues.
6. What kind of methodology that authors follow to extract papers from databases?
7. The small number of papers analyzed in a study can be a major weakness, as it may limit the reliability, validity, and generalizability of the findings. Therefore, it is essential to ensure the appropriate methodology and keywords that have been used to extract data.
8. The authors need to clearly articulate the academic as well as practical implications of this study in a separate section which can be named the theoretical and practical implications of this study.
9. What are some potential implications of these findings for researchers, psychologists, and governments?
If these revisions can be made to the manuscript, I believe that this study can be accepted for publication.
I wish the authors all the very best with this study.
1. Initially, it is important to address the presence of typographical errors and grammatical inaccuracies within the text. Furthermore, the inclusion of lengthy sentences may result in reader confusion and should be carefully considered for improved clarity and comprehension.
Author Response
1. Initially, it is important to address the presence of typographical errors and grammatical inaccuracies within the text. Furthermore, the inclusion of lengthy sentences may result in reader confusion and should be carefully considered for improved clarity and comprehension.
- We hope this version is better
I think the “Introduction” section can be improved. I think the authors should write about consumer behavior (e.g., unconscious and subconscious), which plays a vital role in decision making of consumer toward product. So, I suggest a refs. "exploring global trends and future directions in advertising research: a focus on consumer behavior". In addition, the authors can enrich “introduction” section with neuroscientific tools that can measured the non-verbal interactions and facial expressions, which are benefits in the study, therefore, I suggest the following "neuromarketing tools used in the marketing mix: a systematic literature and future research agenda". I think these references can help you with the issue.
- We included it in the introduction (4th paragraph).
I think the author(s) can add a paragraph at the end of the "Introduction" section about the organizational structure of the paper to ease reading for readers and interested persons. I suggest a reference, which can help you in this issue as well as to improve your literature review 'Scientometric analysis of scientific literature on neuromarketing tools in advertising'.
- We added a paragraph that gives a brief at the end of the introduction (6th paragraph).
The authors should explicitly state the novel contributions of this work and its similarities and differences with previous publications.
- We added at the end of the introduction (6th paragraph).
- The methodology section should be clearly clarified due to the fact that there are some unclear issues.
- We worked on that
What kind of methodology that authors follow to extract papers from databases?
- We kind of followed the PRISMA checklist but we did not state it in the paper since our paper is not a systematic review nor metanalysis.
The small number of papers analyzed in a study can be a major weakness, as it may limit the reliability, validity, and generalizability of the findings. Therefore, it is essential to ensure the appropriate methodology and keywords that have been used to extract data.
- We acknowledged this issue in the study's limitation since the articles we found did not meet our inclusion criteria or assessed different outcomes or the regulations were voluntary.
The authors need to clearly articulate the academic as well as practical implications of this study in a separate section which can be named the theoretical and practical implications of this study.
- We add that in future directions and implications of the study.
What are some potential implications of these findings for researchers, psychologists, and governments?
- We add that in future directions and implications of the study.
Thank you for your constructive comments.
Reviewer 2 Report
The article has improved. Thank you for taking the revisions into account.
I wish the authors good work!
Author Response
Thank you for your constructive comments and support.
Round 3
Reviewer 1 Report
REVIEW COMMENTS
I have only a few concerns about the paper and some suggestions that maybe the authors could consider:
1. Initially, it is important to address the presence of typographical errors and grammatical inaccuracies within the text. Furthermore, the inclusion of lengthy sentences may result in reader confusion and should be carefully considered for improved clarity and comprehension.
2. I The authors have included the same references [18] three times in the same paragraph, as follows: " Advertisers are also using advanced medical technology to their advantage, employing neuroimaging tools such as fMRI, EEG, and fNIRS to analyze the neural responses associated with consumers’ behaviors, such as decision-making, choices, perception, and preferences, that contribute to their reactions to marketing [18]. Visual attention, emotional arousal, and pleasure/displeasure toward a marketing campaign can also be measured using physiological tools such as ET, GSR, and ECG [18]. Both neuroimaging and physiological tools are invaluable in capturing consumers’ mental and physiological responses toward marketing and measuring the effectiveness of food-marketing regulations [18]." I suggest references "exploring global trends and future directions in advertising research: a focus on consumer behavior" and " Biomedical Technology in Studying Consumers’ Subconscious Behavior”, which can be useful in lines 103 and 107, page3.
3. The authors need to clearly articulate the academic as well as practical implications of this study in a separate section which can be named the theoretical and practical implications of this study.
If these revisions can be made to the manuscript, I believe that this study can be accepted for publication.
I wish the authors all the very best with this study.
Initially, it is important to address the presence of typographical errors and grammatical inaccuracies within the text. Furthermore, the inclusion of lengthy sentences may result in reader confusion and should be carefully considered for improved clarity and comprehension.